# SARS-CoV-2 Inactivation in Aerosol by Means of Radiated Microwaves

**DOI:** 10.3390/v15071443

**Published:** 2023-06-27

**Authors:** Antonio Manna, Davide De Forni, Marco Bartocci, Nicola Pasculli, Barbara Poddesu, Florigio Lista, Riccardo De Santis, Donatella Amatore, Giorgia Grilli, Filippo Molinari, Alberto Sangiovanni Vincentelli, Franco Lori

**Affiliations:** 1Elettronica S.p.A., Via Tiburtina Valeria, Km 13.700, 00131 Rome, Italy; 2ViroStatics s.r.l., Viale Umberto I, 46, 07100 Sassari, Italy; 3Defense Institute for Biomedical Sciences, 00184 Rome, Italy; 4Department of EECS, University of California, Berkeley, CA 94720, USA

**Keywords:** SARS-CoV-2, SRET, COVID-19, airborne pathogens, air transmission, microwave inactivation

## Abstract

Coronaviruses are a family of viruses that cause disease in mammals and birds. In humans, coronaviruses cause infections on the respiratory tract that can be fatal. These viruses can cause both mild illnesses such as the common cold and lethal illnesses such as SARS, MERS, and COVID-19. Air transmission represents the principal mode by which people become infected by SARS-CoV-2. To reduce the risks of air transmission of this powerful pathogen, we devised a method of inactivation based on the propagation of electromagnetic waves in the area to be sanitized. We optimized the conditions in a controlled laboratory environment mimicking a natural airborne virus transmission and consistently achieved a 90% (tenfold) reduction of infectivity after a short treatment using a Radio Frequency (RF) wave emission with a power level that is safe for people according to most regulatory agencies, including those in Europe, USA, and Japan. To the best of our knowledge, this is the first time that SARS-CoV-2 has been shown to be inactivated through RF wave emission under conditions compatible with the presence of human beings and animals. Additional in-depth studies are warranted to extend the results to other viruses and to explore the potential implementation of this technology in different environmental conditions.

## 1. Introduction

Airborne pathogens are so called because they are transmitted through the air. There are several lines of evidence to support airborne transmission of SARS-CoV-2, the causative agent of COVID-19 pandemic [1], and air transmission of SARS-CoV-2 has been acknowledged by world health authorities [2,3] based on overwhelming experimental and epidemiological evidences [4,5,6]. Airborne pathogens are also responsible for the transmission of a variety of bacterial, viral, and fungal infections [7,8]. A single human sneeze generates droplets with a size (geometric mean) ranging from 74 to 360 µM [9] that can be spread for several meters [10]. Other respiratory emission events include coughing and talking. It has been shown that 82% of particles detected during tidal breathing have a size of 0.300–0.499 μM [11].

COVID-19 is a highly contagious viral illness that had a catastrophic effect on the world population with well over 750 million cases and almost 7 million deaths worldwide as of 5 June 2023 [12]. A total of 50 vaccines have been approved by at least one country employing several different technology platforms (e.g., mRNA, viral-vector, protein subunit) [13]. A large fraction of people in the world have been vaccinated, with a total of over 13 billion doses of vaccine administered globally as of 5 June 2023 [13] in the attempt to bring the pandemic under control. However, vaccinated people can still be infected by SARS-CoV-2 variants [14]. The grand challenge is therefore to effectively control air transmission of COVID-19 to prevent infection. The virus inactivation technology reported in this manuscript is based on the confined acoustic dipolar mode of resonance with microwaves of the same frequency [15,16]. To this aim, we developed an experimental in vitro model to demonstrate the resonant energy transfer effect of the structure from microwaves to the virus capable of inactivating SARS-CoV-2 in aerosols with a microwave power density that is below the limit for safety allowed by worldwide regulatory agencies. This structure-resonant energy transfer (SRET) effect from electromagnetic waves to confined acoustic vibrations (CAVs) causes the viral membrane fracture through opposite core-shell oscillations [15].

## 2. Background

### 2.1. Microwave-Based SARS-CoV-2 Inactivation

Compared to classical pathogen inactivation, electromagnetic wave radiation can penetrate and heat rapidly and homogeneously, and resonate with microorganisms. In recent years, the applications of electromagnetism for the inactivation of pathogenic viruses have attracted increasing attention [17,18,19].

The viral structure is generally made up of an inside nucleic acid (RNA or DNA) and an external capsid. Nucleic acid determines the genetic and replication properties of the virus. Most emerging viruses of public health concern have an exterior envelope structure consisting of lipids and glycoproteins. Furthermore, envelope proteins determine the specificity of the receptor and act as primary antigens, which the host’s immune system can recognize. An intact structure guarantees the integrity and genetic stability of the virus.

The mechanisms underlying the inactivation of viruses subjected to electromagnetic waves are closely related to the type of virus, and the frequency and power of electromagnetic waves and the viral growth medium are still largely unexplored. Recent studies have focused on the mechanisms of thermal and structural energy transfer for virus inactivation. The thermal inactivation can be attributed to the effects of electromagnetic waves on the water molecules around them. The disadvantage of such a process, compared to the SRET effect via Electromagnetic (EM) wave technique implemented in our work, is the need to generate very high power electric fields, since it has to heat the water up to 100 °C for the destruction of the virus morphology and, therefore, for the reduction of viral activity [20,21]. The core of the technology described in this manuscript was developed by Elettronica SpA (ELT).

### 2.2. Preliminary Tests

The first series of tests was performed in collaboration with the Defense Institute for Biomedical Sciences, Rome, Italy and with the Sacco Civil Hospital, Milan, Italy. The tests were conducted between 2020 and early 2021 in the respective Biosafety Level-3 (BSL-3) laboratories by illuminating viral samples contained in small drops with microwave signals at different frequencies. To check the inactivation ratio, the illuminated viruses were analyzed by plaque assay to measure the residual infectivity of the viruses. The titer of the illuminated viruses was then compared with the titer of the non-illuminated control sets to calculate the inactivation ratio at different frequencies. The initial experiments were performed by using drops larger than those generated by human sneezing.

Subsequently, an innovative method for mimicking the spread of the virus in aerosol has been implemented in cooperation with ViroStatics srl in a controlled and safe way within a BSL-3 laboratory in Alghero, Italy. Virucidal activity has been evaluated against SARS-CoV-2 (original Wuhan strain) in this bioaerosol experimental system according to the methods and test setup described below.

## 3. Methods

### 3.1. Cells and Virus Culture Production

For the experiments conducted at the the Defense Institute for Biomedical Sciences, Rome, Italy (preliminary tests with large drops), a VERO E6 cell line (Cercopithecus aethiops kidney) was purchased from American Type Culture Collection (ATCC) and was cultured in a minimal essential medium (MEM) (Life Technologies, Carlsbad, CA, USA). This was complemented with 10% *v*/*v* heat-inactivated fetal bovine serum (FBS) (Sigma Aldrich, Milan, Italy), 2 mM Glutamine (Life Technologies), and 1% *v*/*v* antibiotic solution (Life Technologies) at 37 °C under 5% CO_2_. SARS-CoV-2, (hCoV-19/Italy/CDG1/2020/EPI_ISL_412973) was isolated from a nasopharyngeal swab by the Department of Infectious Diseases, National Institute of Health, Rome, Italy. The virus was propagated in Vero E6 in 2% FCS until 80% of cellular lysis was observed. The supernatants containing the released viral particles were collected and centrifuged at 600× g for 5 min. The clarified supernatant was frozen and kept at −80 °C until use. Viral titer was determined by plaque assay. Viral culture of SARS-CoV-2 was conducted in the Biosafety Level-3 (BSL-3) facility of the Defense Institute for Biomedical Sciences, Rome, Italy. SARS-CoV-2 cultures and bioaerosol tests were handled in a BSL-3 laboratory at ViroStatics facilities located at the Scientific and Technological Park Porto Conte Ricerche srl (Alghero, Italy). Vero E6 cells were cultured in Dulbecco’s modified Eagle’s medium (DMEM) supplemented with 10% fetal bovine serum (FBS) (Biowest, Nuaillé, France), 1% antibiotic solution penicillin/streptomycin (Biowest, Nuaillé, France), 1% L-glutamine (Biowest, Nuaillé, France), i.e., complete medium, at 37 °C with 5% CO_2_. The human 2019-nCoV strain 2019-nCoV/Italy-INMI1, isolated in Italy (ex-China) from a sample collected on 29 January 2020, was provided by the Istituto Lazzaro Spallanzani, Rome, Italy (Archive E, «Human 2019-nCoV strain 2019-nCoV/Italy-INMI1, clade V», 2020, available on line at https://www.european-virus-archive.com/virus/human-2019-ncov-strain-2019-ncovitaly-inmi1-clade-v (accessed on 20 June 2022)). The virus was propagated in Vero E6 cells as described above to obtain high titer virus (>1 × 10^6^ TCID_50_/mL) and was stored at −80 °C until use. Tissue culture infectious dose (TCID_50_) is defined as the dilution of a virus required to infect 50% of a given cell culture.

### 3.2. Preliminary Tests

To evaluate the antiviral efficacy of electromagnetic wave radiation, a viral suspension of 300 µL SARS-CoV-2 was spotted on a cover glass and illuminated with microwave signals at different frequencies (from 6.5 to 17 GHz). Untreated SARS-CoV-2 was spotted on a cover glass and used as a positive control. The temperature and relative humidity inside the laminar flow cabinet were maintained within a narrow range for testing, specifically 20 ± 4 °C and 19 ± 5%, respectively. After the treatment, a 10-fold serial dilution of the exposed and unexposed viral suspension was inoculated (in duplicate and triplicate) in confluent monolayers of Vero E6 cells seeded in a 12-well plate and incubated for 1 h. The cells were then overlaid with MEM containing 1.5% Tragacanth and 2% FBS (final concentration). The cells were incubated for 72–96 h in a CO_2_ incubator. To calculate plaque forming units (PFU), the cells were washed with sodium chloride (0.9%) physiological solution, followed by staining with a crystal violet solution.

### 3.3. Bioaerosol Tests

The virucidal activity of microwaves produced by a Radio Frequency (RF) generator was evaluated against the SARS-CoV-2 Wuhan strain in a bioaerosol experimental system according to the methods described below. The replicative capacity of the virus treated with microwaves (compared to untreated virus) has been determined in an in vitro system, as follows. A virus containing aerosol was introduced in a plastic, air-proof container (0.5 L volume), and particles with a size up to 10 µm were generated with a commercially available aerosol generator (Omron, Kyoto, Japan)

A viral suspension of the infectious SARS-CoV-2 (original Wuhan strain) was used to generate the aerosol. High titer virus (1 × 10^6^ TCID_50_/mL) was obtained from infected cultures of Vero E6 cells. During the test, the chamber containing the infected aerosol was subjected to irradiation by an electromagnetic signal at the appropriate frequency (that is the speed of the wave or the distance between the start and end of each wave) and amplitude (represented by the height of the wave) to maximize the reduction of the viral load. The RF generator was controlled with a software application installed in a mobile device. The treated aerosol was then recovered by means of active impingement directly in complete medium with 2% serum contained in a special glass collector with an inlet and tangential nozzles, through which air from the plastic chamber was sucked when vacuum was applied (at 12 L/min). Viral particles were absorbed in this medium and then cultured in the laboratory for viral titer determination. The use of this system allowed us to consistently recover 5% of the aerosolized virus. The control virus was processed in the same experimental conditions with the device turned off. The replicative capacity (i.e., viral titer) of the virus treated with microwaves (compared to untreated virus) was determined in an in vitro system. Vero E6 cells were seeded at a density of 20,000 cells/well into a 96-well plate in complete medium at 37 °C and 5% CO_2_. Twenty-four hours after seeding the cells reached ~90% confluency the fetal bovine serum concentration was decreased to 2% to avoid interference with the viral infection. The cells were then infected with 10-fold serial dilutions (8 replicates for each dilution) of the viral suspensions obtained after the impingement procedures for aerosol collection and control. The cells were then cultured for seventy-two hours, and infection was determined by observing the cytopathic effect under an inverted microscope. The viral titer of the recovered virus was determined according to the Reed and Muench method and expressed as TCID_50_/mL [22].

## 4. Results

### 4.1. Analytical Formulation

Viruses can be treated as nano dimensional condensed matter characterized by core-shell charge separations [15]. Considering spherical virions as free homogeneous nanoparticles, we found that the Magnetic Resonance frequencies agree well with that of l = 1 dipolar modes predicted by the elastic continuum theory. The results suggest a modality to set up the microwave apparatus for virus treatment, as discussed later.

Based on Lamb’s theory [23], the frequency of the dipolar spheroidal mode (SPH, l = 1, n = 0) for any spherical particle can be estimated using the following eigenvalue equation:(1)$$4\frac{j_{2}\left(\xi \right)}{j_{1}\left(\xi \right)}\xi -\eta^{2}+2\frac{j_{2}\left(\eta \right)}{j_{1}\left(\eta \right)}\eta =0$$
where *ξ* = 2*πν*R/*V_L_*, *η* = 2*πν*R/*V_T_* and *j_l_*(*η*) is the spherical Bessel function of the first kind of the *l*-th order, *R* denotes the radius of the particle, and *V_L_* and *V_T_* are, respectively, the sound velocities of longitudinal and transverse waves.

Since most typical condensed matters have around 0.3 Poisson’s ratio, the *V_L_*/*V_T_* ratio remains approximately 2 [24]. Most viruses are composed of lipids, proteins, and genomes. The *V_L_* of lipid is ~1500 m/s [25] and most protein crystals’ *V_L_* are around 1800 m/s [26]. The reported *V_L_* of the uncompressed wet genomes is ~1700 m/s [27]. Since most emerging viruses of public health concern have highly compressed genomes, and their capsid proteins have strong tension, the effective *V_L_* should be in the order of or more than 1500–1800 m/s if the virus does not have a lipid envelope.

Considering the previous observation, for any variable *z* and any integer *m* > 0:(2)$$\mathrm{when}\;z\;\to \;0,\;j_{m}\left(z\right)≃\frac{1}{2^{m}m!}z^{m}$$

Therefore, (*R/V_L_*) → 0 and (*R/V_T_*) → 0; *ξ* → 0, *η* → 0 and the eigen value Equation (1) becomes:(3)$$4\frac{\xi }{4}-\eta^{2}+2\frac{\eta }{4}\eta =0$$

Consequently:(4)$$\eta^{2}=2\xi$$
since *V_L_*/*V_T_* = 2 → *η* = 2*ξ*, which yields (with Equation (3)) 2*ξ* = 1.

In conclusion, since the angular frequency is ω = 2*πν*, the resonant frequency *f* is
(5)$$f=\frac{V_{L}}{2R}$$

For the analytical estimation of *f*, the correct estimation of *V_L_* is fundamental, while it is possible to measure the radius R of the virions with high accuracy.

Since the range of the radius of the SARS-CoV-2 particle is between 60 nm and 140 nm [28], the resonant frequency was first estimated to be in a range from 6.5 GHz to 16 GHz. To support the logic of this estimation, we used the theoretical model and related experiments reported in [29], identifying the confined acoustic vibration that can cause strong resonant microwave absorption for the rod-shaped White Spot Syndrome Virus (WSSV) virion.

Since the specific core-shell space charge separation for resonance dipolar interaction excitation is valid regardless of the virus shape and the confined “dipolar-like” acoustic phonon modes, we expect that a similar approach can be used for the de-activation of different types of viruses and even bacteria. For example, we believe that the charge distribution for the non-enveloped virus containing only two elements, which are the genome (RNA or DNA) and the virus-encoded protein capsid surrounding the genome [30], can be modelled in the same way to achieve inactivation by MRA.

### 4.2. Bioaerosol Tests Setup

To perform the inactivation tests on pathogens, a demonstrator was developed whose transfer function is that of a synthesized RF generator.

The unit consists of an Ultra-Wideband (UWB) frequency-tunable synthesizer operating from the C band to the Ku band, a medium power and a high-power microwave amplifier, and a digital variable attenuator to generate as output a microwave signal that is variable in frequency and amplitude (Figure 1).

To set all the possible configurations of the RF components, an embedded software written in C++ was developed on the ESP32 platform using Visual Studio Code (https://code.visualstudio.com/ (accessed on 20 June 2022)).

The final power amplifier of the demonstrator was developed using the latest 0.15 µm GaN on SiC solid-state High-Electron-Mobility Transistor technology and can deliver up to 10 W in a UWB.

The RF output of the transmitter was connected via an RF cable to a horn antenna capable of generating an appropriate electromagnetic field near the area to be sanitized. During the test, the functionality of the treatment was checked by means of a Vivaldi UWB test antenna connected to a spectrum analyzer (Figure 2a).

The system has an *EIRP* [31] (Effective Isotropic Radiated Power) that is a function of frequency since the antenna horn, cable, and power amplifier have frequency dependent antenna gain, insertion loss, and output power, respectively. Assuming that the distance between the antenna and the base where the samples are placed is in the far field of the antenna, we obtain the following equation for the *EIRP* expressed in *dBmi* [31]:(6)$${\left.EIRP\left(f\right)\right|}_{dBmi}={\left.Pout\left(f\right)\right|}_{dBm}+{\left.Gant\left(f\right)\right|}_{dBi}-{\left.IL\left(f\right)\right|}_{dB}$$
where *Pout*(*f*) is the RF amplifier output power in *dBm*, *Gant*(*f*) is the antenna gain in *dBi*, and *IL*(*f*) is the cable insertion loss in *dB*.

Since the distance between the samples and the antenna is approximately 20 cm, the power density in proximity of the sample is:(7)$${\left.S\left(f\right)\right|}_{W}=\frac{{\left.EIRP\left(f\right)\right|}_{W}}{4*\pi *{{\left.R\right|}_{m}}^{2}}$$
where *R* is the distance in meters between the antenna and the base where the sample is placed. Moreover:(8)$${\left.S\left(f\right)\right|}_{W}=\frac{{\left.{\left|E(f)\right|}^{2}\right|}_{V/m}}{2*{\left.Z\right|}_{ohm}}$$
where *E*(*f*) is the peak value of the continuous wave (CW) sinusoidal electrical field and *Z* is the impedance of free space.

Using Equations (7) and (8), given *EIRP*(*f*) and *R*, we can express the peak of the electrical field as a function of frequency.

Before starting the treatment, an appropriate calibration of the test setup, modifying the gain of the amplifier, and, consequently, the Pout to obtain the same values of the electrical field for all the frequency, was carried out to evaluate the correct inactivation of the virus for each specific value of the electric field intensity associated with each specific frequency. For this purpose, a broadband field meter was used (Figure 2b). The aim of this calibration phase is to have the same electrical field for all the selected frequencies for each specific treatment.

### 4.3. Bioaerosol Tests Outcome

To evaluate the antiviral efficacy of the resonant effect, we first measured the residual viral infectivity of SARS-CoV-2 after illumination with microwaves of different frequencies. Each identical viral sample was placed below the horn antenna and irradiated with a fixed frequency and high-level amplitude electromagnetic field (i.e., 400 V/m). The radiated frequency is different among all the tests (i.e., with 1 GHz step except 6.5 GHz due to RF synthesizer limitations) with the aim to measure different levels of the inactivation ratio and, consequently, to detect the resonant frequency (i.e., where the maximum of inactivation is achieved considering a fixed amplitude of the electric field). The inactivation ratio (100 − (N/N0 × 100)) was calculated by comparison of the PFU count of the illuminated viruses (N) and the unilluminated control sets (N0) at different frequencies (6.5–17 GHz).

As summarized in Figure 3, an inactivation ratio higher than 40% was observed in a frequency range of 9 GHz to 12 GHz with a peak 65.9% inactivation ratio located at the resonant frequency of 10 GHz.

These initial tests were performed with large size DMEM drops (300 µL), larger than those commonly spread during sneezing, coughing, or talking [9,11] laid out on a solid surface to obtain an initial indication of preferred frequency (i.e., the resonant frequency that correspond to the maximum of the inactivation ratio). For these tests, a high intensity radiated electric field was required because of the strong reduction of intensity of the field inside the drop, due to the reflection caused by the difference in the dielectric constant between air and DMEM (DMEM relative permittivity -ε_r_- being ca. 80) and the absorption caused from the DMEM drops.

Since the viral drop size influences the inactivation capacity of microwaves, we decided to test the antiviral efficacy of the RF generator in a more physio-pathological relevant aerosol system, selecting a range of 8–10 GHz based on the previous experiments and on preliminary aerosol testing.

First, we used a high electromagnetic field (400 V/m) to confirm viral inactivation in the aerosol system. Then, the emitted power was gradually reduced to 6 V/m, a level that is compatible with the presence of humans [32], without the loss of antiviral activity (Figure 4a). A further decrease in potency (e.g., 3 V/m) resulted in suboptimal virus inactivation (Figure 4b).

The results presented in Figure 4 were obtained using the entire frequency band (i.e., 8–10 GHz). In fact, if we consider the dimensional spread of the virions, and the inversely proportional relation between virion radius and resonance frequency, using a frequency band instead of a single frequency produces a higher ratio of inactivation. To span the frequency band of interest, we use a set of N discrete frequencies that are separated by a uniform step size. At each frequency, we expose the aerosolized virus for a period of time (i.e., the dwell time DT). To minimize the total time of exposure, we minimized the product N × DT keeping the inactivation ratio constant.

The viral inactivation ratio at a different value of frequency step and frequency dwell time are reported in Figure 5 and Figure 6, respectively. The frequency step could be increased up to 20 MHz (Figure 5a) while a further increase to 40 MHz resulted in a loss of activity (Figure 5b).

The dwell time could be reduced to 0.05 s, but this time had to be constant. Interrupting the emission (e.g., 0.5 s on and 0.5 s off) before moving to the next frequency step resulted in a loss of activity.

By applying an optimized frequency step of 20 MHz and an optimized constant dwell time of 0.05 s (but not under unoptimized conditions, which are not shown), the total time exposure could be reduced to one minute without a loss in antiviral activity (Figure 7).

Table 1 summarizes all aerosol-based SARS-CoV-2 inactivation experiments, under optimized conditions, as they appeared sequentially in Figure 5, Figure 6 and Figure 7, that is, in different experimental conditions (regarding potency, step, dwell time, and time of exposure). Overall, compared to the untreated control, the average SARS-CoV-2 inactivation was over 90%, with a small (5.53) Standard Deviation (SD) reflecting marginal differences among the experiments.

## 5. Discussion

The inhalation of aerosol and droplets from infected individuals is recognized as the primary mode of respiratory virus transmission, with droplet dispersion influenced by size and environment.

This study allowed the identification of the microwave resonant frequencies for SARS-CoV-2 and the demonstration of a reproducible virucidal effect using an RF generator. Our preliminary results, obtained in a controlled laboratory environment mimicking a natural airborne virus transmission with aerosol particles up to 10 μm in diameter, demonstrate for the first time that the viability and infectivity of SARS-CoV-2 can be reduced by 90% (tenfold) through RF wave emission. The application of this technology has the potential to aid in the reduction of virus transmission by sanitizing indoor environments where virus-laden airway fluid (greater than 5μ m in diameter) or aerosol (smaller than 5 μm) can remain for hours [16].

This virucidal effect could be achieved by using an RF-wave emission system with a power level that is safe for people according to the Specific Absorption Rate (SAR) US, Canadian, European, and Japanese standards and specifications, thus offering an important method for human environment sanitization.

Additional studies will be needed to confirm the results in different environmental conditions, and to demonstrate the efficacy of microwave inactivation for airborne viruses other than SARS-CoV-2, with the intent of building a library of frequencies that are effective for different viruses.

## Figures and Tables

**Figure 1 viruses-15-01443-f001:**
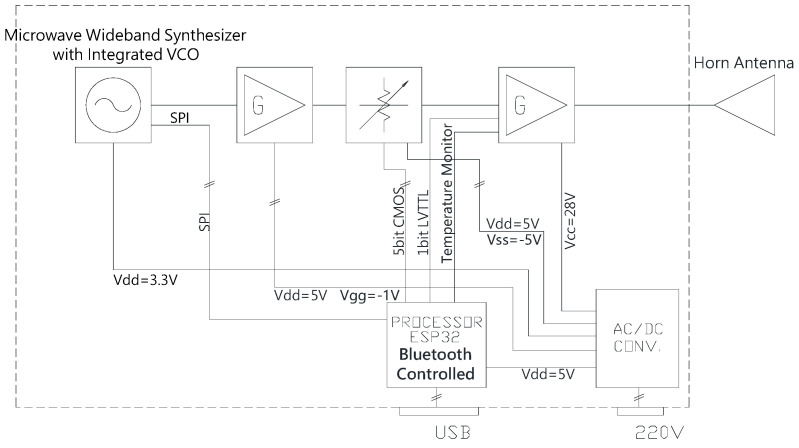
RF Generator Block Diagram. Parts of the demonstrator: ultra-wideband frequency tunable synthesizer, a medium power, and high-power microwave amplifier, and a digital variable attenuator generating the microwave signal with predetermined frequency and amplitude.

**Figure 2 viruses-15-01443-f002:**
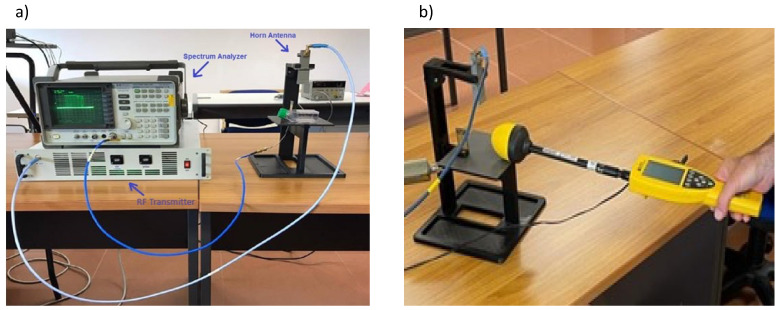
(**a**) RF test set. RF transmitter was connected to the horn antenna generating the electromagnetic field over the area containing the pathogen. A spectrum analyzer connected to a Vivaldi UWB antenna was employed to verify the appropriate emission of microwaves during treatment. (**b**) Electromagnetic field calibration. A calibration of the test setup was performed before each experiment by means of a broadband field meter to properly assess the inactivation potency of the electromagnetic field at a specific frequency.

**Figure 3 viruses-15-01443-f003:**
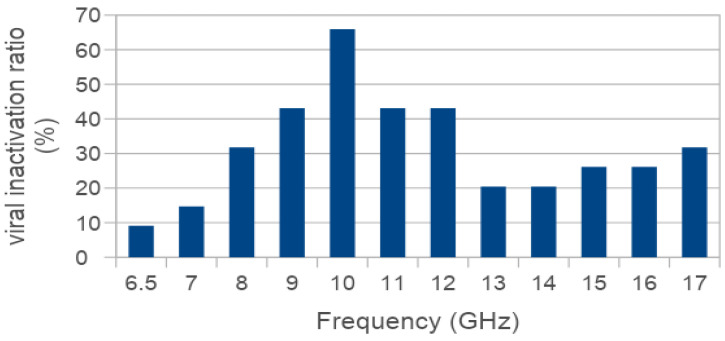
Inactivation ratio of SARS-CoV-2 exposed to illuminating microwaves at different frequencies. Viral samples in 300 µL drops laid on a solid surface were irradiated with microwave signals at different frequencies and high levels of electromagnetic field (up to 400 V/m). Residual viral infectivity after exposure to microwaves was determined by calculating the ratio of the plaque forming unit (PFU) count obtained for the illuminated viruses and the unilluminated control set. A higher than 40% inactivation ratio was observed in the frequency range 9–12 GHz.

**Figure 4 viruses-15-01443-f004:**
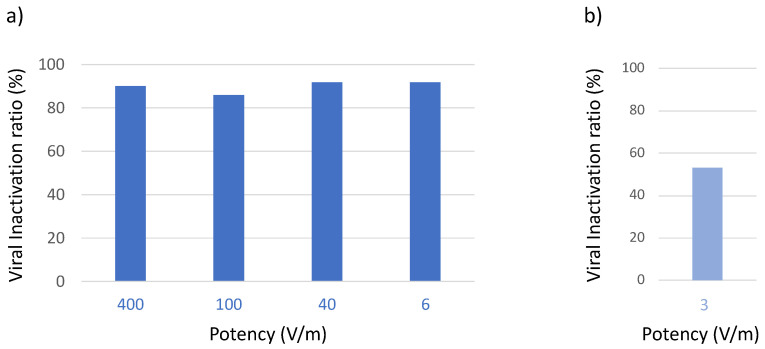
Effect of potency of the electromagnetic field on antiviral activity in the bioaerosol system. Aerosolized virus was exposed to microwaves of different potency in the range 8–10 GHz. Residual viral infectivity after exposure to microwaves was determined by calculating the ratio of the viral titer obtained for the illuminated virus aerosol and the viral titer of the unilluminated control set. Viral titer was determined with the Reed and Muench method through observation of the cytopathic effect of the virus in Vero E6 cells. (**a**) Virucidal activity was maintained when potency was gradually reduced to 6 V/m (optimized conditions). (**b**) Virucidal activity was reduced when potency was set to 3 V/m.

**Figure 5 viruses-15-01443-f005:**
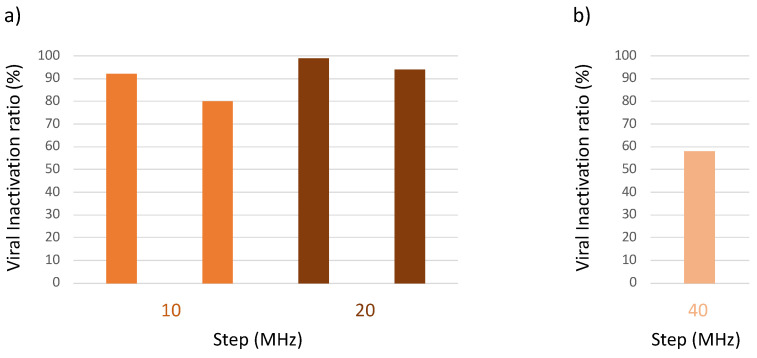
Effect of frequency step size on antiviral activity in the bioaerosol system. The aerosolized virus was exposed to microwaves of different frequency step size in the range 8–10 GHz. Residual viral infectivity after exposure to microwaves was determined by calculating the ratio of the viral titer obtained for the illuminated virus aerosol and the viral titer of the unilluminated control set. Viral titer was determined with the Reed and Muench method through observation of the cytopathic effect of the virus in Vero E6 cells. (**a**) Virucidal activity was maintained when the frequency step size was increased from 10 to 20 MHz (optimized conditions). (**b**) There was a loss of activity when the frequency step size was increased to 40 MHz.

**Figure 6 viruses-15-01443-f006:**
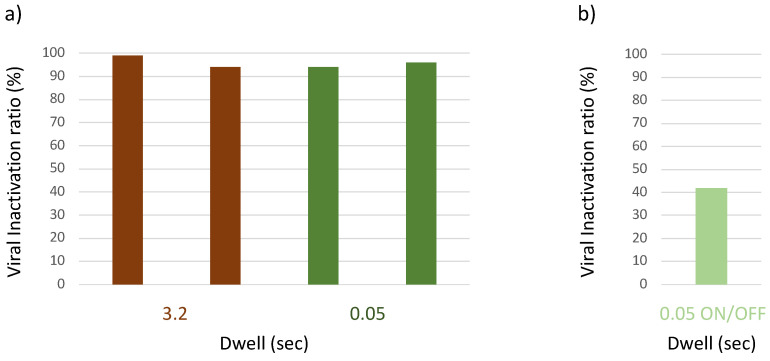
Effect of dwell time on antiviral activity in the bioaerosol system. The aerosolized virus was exposed to microwaves of different dwell time in the range 8–10 GHz. Residual viral infectivity after exposure to microwaves was determined by calculating the ratio of the viral titer obtained for the illuminated virus aerosol and the viral titer of the unilluminated control set. Viral titer was determined with the Reed and Muench method through observation of the cytopathic effect of the virus in Vero E6 cells. (**a**) Virucidal activity was maintained when dwell time was reduced 3.2 to 0.05 s (optimized conditions). (**b**) Interrupting the emission during the dwell resulted in loss of activity.

**Figure 7 viruses-15-01443-f007:**
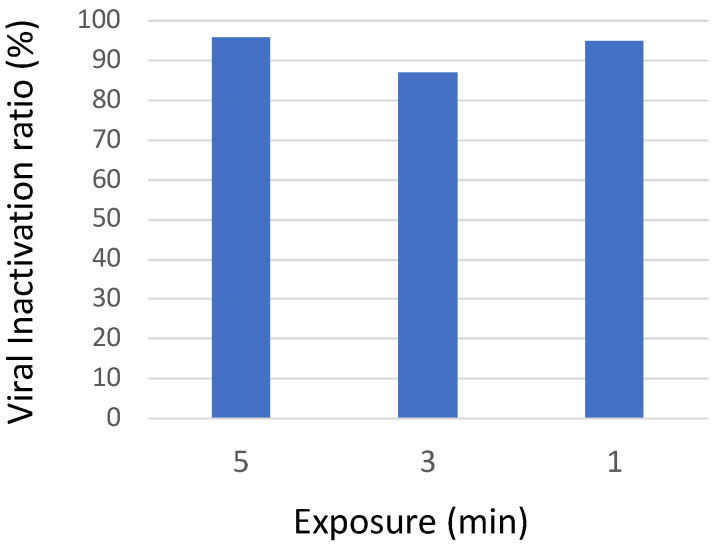
Effect of different time exposures on antiviral activity in the bioaerosol system. The aerosolized virus was exposed to microwaves for different treatment times in the range 8–10 GHz. Residual viral infectivity after exposure to microwaves was determined by calculating the ratio of the viral titer obtained for the illuminated virus aerosol and the viral titer of the unilluminated control set. Viral titer was determined with the Reed and Muench method through observation of the cytopathic effect of the virus in Vero E6 cells. Virucidal activity was maintained when exposure time was reduced to 1 min.

**Table 1 viruses-15-01443-t001:** SARS-CoV-2 inactivation under optimized conditions. Summary of viral titer obtained for untreated control and aerosolized virus exposed to microwaves in the range 8–10 GHz from the experiments described in Figure 4a, Figure 5a, Figure 6a and Figure 7.

Experiment Number	Titer *w*/*o* Treatment	Inactivation (%)
	Titer with Treatment	
1	3.98 × 10^3^	90.00
	3.98 × 10^2^	
2	3.98 × 10^3^	85.88
	5.62 × 10^2^	
3	3.98 × 10^3^	92.06
	3.16 × 10^2^	
4	3.98 × 10^3^	92.06
	3.16 × 10^2^	
5	1.74 × 10^3^	79.37
	3.59 × 10^2^	
6	5.13 × 10^4^	98.77
	6.31 × 10^2^	
7	1.29 × 10^5^	93.84
	7.94 × 10^3^	
8	1.29 × 10^5^	93.84
	7.94 × 10^3^	
9	1.29 × 10^5^	96.40
	4.64 × 10^3^	
10	1.20 × 10^5^	86.83
	1.58 × 10^4^	
11	1.20 × 10^5^	95.32
	5.62 × 10^3^	
	**Inactivation (Average)**	**91.31**
	**Inactivation (SD)**	**5.53**

## Data Availability

Data is contained within the article.

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
