# Peer review of "SARS-CoV-2 Inactivation in Aerosol by Means of Radiated Microwaves"

_viruses, 2023, doi:10.3390/v15071443_

Round 1
Reviewer 1 Report
Comments and Suggestions for Authors
Technologies to sanitize indoor air of pathogens such as SARS-CoV-2, while humans occupy those spaces, are needed. This article describes the use of radio frequency irradiation as such a technology. As the authors state, this is a novel technology for this purpose. I am not aware of previous reports on this topic, although I have not searched the literature to confirm this. I presume that the authors have done so.
I suggest that you adopt the term air or airborne instead of aerosol. There is a debate raging in the field over what the primary transmission mode for SARS-CoV-2 and other respiratory viruses is. There is no need for you to take a side on this question. You are describing a method to inactivate viruses suspended in the air. I would stay on this neutral ground. You have done this in the beginning of your introduction. I suggest retaining this language throughout, except in cases such as your title and experiments, where an aerosol was generated and tested.
Some specific suggestions:
line 12. Not sure class is the right term. how about family
line 65. I would change most to many
line 125 to 128. Line 128 refers to a chamber. This should be described. Was the negative control also applied to a cover glass and kept in the chamber? This should be described in more detail. The h in Ghz should be capitalized.
line 135. the physiological solution should be better defined
line 170. Reed and Muench should be cited in the references
Section 4.1. As I was reading this, a couple of questions came to mind: 1) will this technology only work with spherical viruses such as SARS-CoV-2 (and other coronaviruses)? Will it work with elongated viruses such as filoviruses or poxviruses? 2) will this technology work for non-enveloped viruses? I think you have partially answered this question in Discussion line 350, but this should be expanded on with reference to the questions posed above.
Line 186. Again, I do not feel it is correct to say that most viruses are [enveloped]. I would say many viruses, or you could say most emerging viruses of public health concern.
line 191. is soft the right term? is it better to say lipid?
line 232. I suggest sanitized rather than deactivated
line 246. I would replace set with assess
Section 4.3. One thing is not clear to me, and therefore may not be clear to your readers who may not be experts in RF dosimetry. The units GHz define the frequency of the EM radiation. Is there not a need to define a contact time with the radiation in Figures 6 and 7? What is meant by amplitude (line 268)? This needs to be clarified. How does RF potency in V/m relate to the applied frequency? Do these two terms combine in some manner to establish an applied RF dose. Can you explain for the readers' benefit.
Line 292. Frequency step size needs to be explained. Frequency dwell time also could use explaining. I had assumed this meant exposure time (contact time), but I see this is discussed below separately.
Line 312.
Most physical inactivation modalities (heat, UV, gamma, electron beam) display approximate first-order kinetics for inactivation vs. time. It is curious that you are seeing essential similar results for 3.2 seconds vs. 0.05 seconds dwell time, and with 1 vs. 9 minutes exposure time. Can this be explained?
As a general comment, you have not addressed the point that a 1 log10 reduction in virus titer (i.e., 90%) is marginally effective. In the biopharmaceutical industry, a purification step displaying < 1 log10 reduction would be considered ineffective. In the disinfectant world, a reduction of 3 logs would be expected. The failure to show a dose-response relationship in terms of irradiation potency, dwell time, or exposure time is also disconcerting. This implies that increasing one of these parameters might not help you improve efficacy. This should be discussed in a revision.
Comments on the Quality of English Language
I have suggested a few changes in terminology. Otherwise I find the English language understandable and sufficient.
Reviewer 2 Report
Comments and Suggestions for Authors
The manuscript viruses-2417805 entitled SARS-CoV-2 Inactivation in Aerosol by means of Radiated Microwaves by Antonio Manna and coworkers studied a method of inactivation based on the propagation of electromagnetic waves in the area to be sanitized.
They consistently achieved a 90% (tenfold) reduction of infectivity, including after short treatment using a Radio Frequency (RF) wave emission with a power level safe for people according to most regulatory agencies including Europe, USA and Japan.
Experimental design is clear and discussion is supported by results.
The overall merit is average.
Minor comments
Line 41,43 and 45 some references are missing.
Line 95 the title should be with the following text in the following page.
Line 181 and followings: the mathematical formula should be in line with the text
Figure 2 in not informative and could be eliminated.
Figure 4 and figure 5 are very little informative and could be eliminated or unified.
Comments on the Quality of English LanguageThe english lenguage is good.
Round 2
Reviewer 1 Report
Comments and Suggestions for Authors
I am happy with your arguments about the tenfold reduction in virus load potentially having impact. We will allow the readers to decide whether this is significant enough.
I believe that you added the phrase most 187 emerging viruses of public health concern to the wrong instance of "most viruses" (i.e., on new line 187). This belongs on line 185 and not on line 187. If you agree, this could be corrected in a final revision, or could be corrected at the galley proof stage.